# Microbiological Aspects of Pharmaceutical Manufacturing of Adipose-Derived Stem Cell-Based Medicinal Products

**DOI:** 10.3390/cells12050680

**Published:** 2023-02-21

**Authors:** Ilona Szabłowska-Gadomska, Monika Humięcka, Joanna Brzezicka, Anna Chróścicka, Joanna Płaczkowska, Tomasz Ołdak, Malgorzata Lewandowska-Szumiel

**Affiliations:** 1Laboratory for Cell Research and Application, Center for Preclinical Research and Technology, Medical University of Warsaw, Banacha 1b, 02-097 Warsaw, Poland; 2BBMRI.pl Consortium, 61 Żwirki i Wigury Street, 02-091 Warsaw, Poland; 3Department of Histology and Embryology, Medical University of Warsaw, Chałubińskiego 5, 02-004 Warsaw, Poland; 4Polish Stem Cell Bank (PBKM), Jana Pawła II 29, 00-867 Warsaw, Poland

**Keywords:** microbiological control, mesenchymal stem cell, cell therapy, ATMP, GMP, ADSC

## Abstract

Subcutaneous adipose tissue is an excellent source of mesenchymal stem cells (ADSCs), which can be used in cell therapies as an active substance in advanced therapy medicinal products (ATMPs). Because of the short shelf-life of ATMPs and the time needed to obtain the results of microbiological analysis, the final product is often administered to the patient before sterility is confirmed. Because the tissue used for cell isolation is not sterilized to maintain cell viability, controlling and ensuring microbiological purity at all stages of production is crucial. This study presents the results of monitoring the contamination incidence during ADSC-based ATMP manufacturing over two years. It was found that more than 40% of lipoaspirates were contaminated with thirteen different microorganisms, which were identified as being physiological flora from human skin. Such contamination was successfully eliminated from the final ATMPs through the implementation of additional microbiological monitoring and decontamination steps at various stages of production. Environmental monitoring revealed incidental bacterial or fungal growth, which did not result in any product contamination and was reduced thanks to an effective quality assurance system. To conclude, the tissue used for ADSC-based ATMP manufacturing should be considered contaminated; therefore, good manufacturing practices specific to this type of product must be elaborated and implemented by the manufacturer and the clinic in order to obtain a sterile product.

## 1. Introduction

After years of the successful use of bone marrow hematopoietic stem cell transplantation, the discovery of nonhematopoietic stem cells in the bone marrow by Friedenstein opened the way to completely new cell therapies [1]. These cells were named mesenchymal stem cells (MSCs), as proposed by Caplan [2]. Although this term is currently under discussion and many authors, including Caplan himself, generally use the term “stromal” instead of “stem” cells, the MSC abbreviation is clearly recognizable and for the purpose of this manuscript, the classic name will be used.

Over the last three decades, it has been proven that it is possible to obtain MSCs from various tissues, such as the umbilical cord, dental pulp, dermis, peripheral blood, and adipose tissue [3,4,5,6,7,8]. The latter is currently widely exploited as an MSC source due to its easy availability and MSC abundance, which is 100-fold higher than that of bone marrow [6,9,10]. Although MSCs obtained from different sources are not exactly the same, they are of interest as an active substance in a wide variety of medical applications. Due to their ability to differentiate toward various phenotypes, there was initially great interest in the possibility of their use for the regeneration of various tissues [11,12]. In addition, there has recently been growing evidence of their immunomodulatory properties, which suggests possible MSC applications in healing wounds in an inflammatory environment, autoimmunological diseases, and graft versus host disease (GvHD), for example [13,14,15,16,17].

There is a rich and continuously growing index in the Clinicaltrials.gov database (https://clinicaltrials.gov) which is also reflected in a number of review papers [3,13,14,16,18]. The road toward MSC-based products becoming available on the medical market is demanding and time- and cost-consuming [19]. This is because, unlike bone marrow transplants, they are considered to be medicinal products in the vast majority of cases. As such, they must fulfill not only transplantological requirements but also fall under the regulations of pharmaceutical laws, requiring centralized marketing authorization [3,19,20]. Currently, there are 54 approved ATMPs on the global medical market, excluding genetically modified products [20]. Currently, ten MSC-based ATMPs have achieved market approval [20,21]. Among them, there is Alofisel, approved for use by the EU, and in the US there is Prochymal, renamed to Remestemcel-L, already approved for use in Canada and New Zealand, which is under FDA consideration as a GVHD treatment for children [22,23]. The time gap between clinical trials and market approval is understandable in view of the regulatory pathway for ATMPs [19,24]. Although the regulatory details may differ in particular countries, the general rule is similar [20,25]. In the EU, there is the Regulation of the European Parliament and of the Council of 2007, which introduced the term Advanced Therapy Medicinal Products (ATMP) [26]. Apart from the details, products containing such cells as MSCs are not considered transplants, but rather ATMPs, if they are substantially manipulated or if their intended essential role in the recipient is not the same as in the donor [26,27]. In particular, cell culture or cell isolation via the enzymatic digestion of tissue are considered to constitute substantial manipulation [27]. Therefore, all steps, from cell isolation to product preparation, must be performed following good clinical manufacturing practice (GMP) regimens in accordance with pharmaceutical law. Such rules apply not only to products with market approval, but also to advanced therapy investigational medicinal products (ATIMP), i.e., those which are prepared for clinical trials.

Thus, the further development of cell-based therapies is inseparable from the distinctive manufacturing requirements of ATMPs, and the exchange of experience in this area represents an area of continued progress. This paper is focused on clinical microbiology laboratory practice in terms of MSC-based ATIMP production. Regardless of the fact that the general GMP rules for medicinal products must be respected, there are some distinctive issues relating to products containing cells collected from culture that should be considered in terms of sterility. Firstly, the starting material, i.e., the human tissue used for cell isolation, may be contaminated. Secondly, all of the manufacturing steps, including many activities typical to cell culture and characterization, are performed in a manner that keeps the cells alive. Finally, for non-cryopreserved cells from culture, the results of the sterility test of the product are not available upon release, which is accepted by the regulatory bodies [28]; however, this imposes the need for special attention and control during all stages of production.

This study is based on original data taken from a two-year-long continuous production of ADSC-based ATIMPs (advanced therapy investigational medicinal products), i.e., ATMPs manufactured for the purpose of a clinical trial. It presents well-documented data illustrating the microbiological characteristics of the manufacturing process controlled during various stages over the whole road of the product from the clinic to the lab and back in order to share the experience, analyze the critical points, and propose steps towards improving ATIMP manufacturing.

## 2. Materials and Methods

### 2.1. Sterility Testing of the Biological Material

Lipoaspirates were acquired via standard liposuction from 100 adult patients at two aesthetic clinics in Poland during a two-year clinical trial after obtaining their informed consent (https://www.clinicaltrialsregister.eu EudraCT Number: 2016-004110-10). Prior to digestion, the adipose tissue was washed extensively with a 1% antibiotic-antimycotic solution containing 10,000 IU penicillin, 10,000 μg/mL streptomycin and 25 μg/mL amphotericin B (Corning, Manassas, VA, USA) in phosphate-buffered saline (PBS) (Thermo Fisher Scientific, Bleiswijk, The Netherlands). The tissue was then digested with collagenase NB 6 GMP Grade (SERVA Electrophoresis GmbH, Heidelberg, Germany) and reconstituted in PBS (Thermo Fisher Scientific, Bleiswijk, The Netherlands). After digestion, the collagenase was inactivated. The stromal vascular fraction (SVF) was separated by centrifugation (350× *g*/10 min/22 °C) and filtration (strainer 100 µm) and washed with sodium chloratum (Fresenius Kabi, Warsaw, Poland). Depending on the type of product, the ATMPs were either prepared (SVF) or further cultured (ADSCs). To prepare the ADSC product, the cells were plated in T75 culture flasks and cultured at 37 °C and 5% CO_2_ in a humidified atmosphere in the complete GMP-grade culture medium: a MSC NutriStem^®^ XF Basal Medium with Supplement Mix (Biological Industries, Beit Haemek, Israel) and antibiotic-antimycotic solution (Corning, Manassas, VA, USA). The latter contained 10,000 IU penicillin (class Penicillins β-lactam antibiotics, 10,000 μg/mL streptomycin (aminoglycoside antibiotics class) and 25 μg/mL amphotericin B (polyene antifungal antibiotic class), and its final concentration in the medium (either PBS, for washing, or full culture medium) was equal to 0.1% (Figure 1).

The presence of microbial contamination was checked at five different stages during the cell harvesting process, including Stage 1: where the samples were taken during the preparation of tumescent fluid before liposuction (fresh tumescent fluid, which is routinely used to ensure a painless and relatively bloodless liposuction procedure—contains 40 mL of 1% lidocaine and 2 mg adrenaline in 1000 mL lactated Ringer’s solution); Stage 2: where the material taken after liposuction (lipoaspirate); Stage 3: where the samples were collected during the intermediate stage of processing (the medium from the isolated cells); and Stage 4: the final stage of isolation (where the medium was separated from the last wash of the cells). The samples from the first and second stages were taken in the clinic, and the samples from the third and fourth stages were taken in the laboratory. Additional samples were also taken during the in vitro cell culture process (cell culture supernatant—Stage 5). The examination of the samples for the presence of microbial contamination was performed using a BACTEC^TM^ automated system. The use of the BD Bactec system has been validated according to the EU Pharmacopoeia (Polish version) following the scheme for alternative methods [29]. For the validation of Bactec method according to the Pharmacopeia [29], the following strains were used: *Pseudomonas aeruginosa ATCC9027*; *Bacillus subtilis ATCC6633*; *Staphylococcus aureus ATCC6538*; *Candida albicans ATCC 10231*; *Bacteroides fragilis ATCC25285*; *Clostridium sporogenes ATCC19404*; *Streptococcus pyogenes ATCC19615*; *Aspergillus brasiliensis (niger) ATCC16404*; and *Propionibacterium acnes ATCC11827*. Additionally, microorganisms were inoculated in the solution in which the cells were previously suspended (e.g., culture medium, medium with 0.1% AAS). The samples prepared in this way were then tested. The applied methodology has been positively verified by the Polish Pharmaceutical Regulatory Authority. BACTEC^TM^ Plus Aerobic/F Culture Vials and BACTEC^TM^ Plus Anaerobic/F Culture Vials media are enriched soybean-casein digest broths with an antibiotic-removing resin (Becton Dickinson and Company, Sparks, NV, USA). The BACTEC^TM^ Plus Aerobic/F culture vial medium contains CO_2_, and the BACTEC^TM^ Plus anaerobic/F culture vial medium is dispensed with CO_2_ and N_2_ [30]. The bottles were inoculated with the samples under a laminar air flow chamber (LabGard Class II, Type A2 Biosafety Cabinet) and incubated at 35 °C ± 2.5 for 14 days. In the case of a positive signal, the identification of the contamination was conducted and the results were collected.

In the final stage of production, the endotoxin levels were determined in the supernatant sample after the last centrifugation during the process of washing cells with sodium chloratum (see Figure 1). For this purpose, the Endosafe^®^ endotoxin test system was used. The acceptable endotoxins level was <2 EU/mL (according to Polish Pharmacopeia chapter 5.1.10) [31].

When positive microbiological test results of the starting material—lipoaspirate (e.g., after 24 h), were obtained, corrective actions were implemented, e.g., extending the culture time, changing the number of passages, or extra freezing/thawing steps. These steps were carried out until a sterile microbiology test result was obtained from at least two of the subsequent probes of the cellular material processing.

The sampling points of the microbiological and endotoxin tests are indicated in Figure 1, illustrating the subsequent stages of the procedure, from the collection of the starting material to the production of ATIMP (Figure 1A) and the detailing of the manufacturing steps used in the laboratory (Figure 1B).

### 2.2. Environmental Monitoring of Microbiological Quality

Three methods were used to monitor the microbiological safety of the manufacturing environment: active air sampling (volumetric sampling), passive air sampling (settle plates), and surface sampling (contact plates). All of the samples were collected according to an approved microbiological monitoring program and at a frequency recorded in accordance with the quality management system that was developed according to the legal requirements of the pharmaceutical quality assurance system.

This program includes the monitoring of the cleanliness of laboratory rooms and consists of checking for the presence of microorganisms as well as the content of inanimate particles in the air. The sampling sites, their number, and their frequency were also determined. The activities carried out ensured that an appropriate level of cleanliness was maintained in the production environment, minimizing the risk of the product becoming contaminated with microorganisms.

For the volumetric sampling, a MICROFLOWα 90/C sampler (AQUARIA, Lacchiarella, Italy) was used. The aspirated volume was 1 m^3^, and the air was collected every 10 min on 90 mm Petri dishes (IRR Tryptone Soya 1.6% Agar + Neutralizer N° 4, Redipor, Bicester, UK). The test was carried out every 3 months. Passive air sampling was conducted using 90 mm Petri settle plates (IRR Tryptone Soya 1.6% Agar + Neutralizer N° 4, Redipor, Bicester, UK). Open plates were exposed to air (including the laminar airflow chambers) for a maximum of 4 h to prevent agar desiccation. In Grade A and B areas, the test was always carried out during all stages of manufacturing. In Grade C and lower areas, the test was carried out at least every two weeks.

While conducting the experiment, the frequency of sampling in specific places and purity classes was reduced. Changes were made on the basis of a systematic analysis of the trends in the microbiological results and after conducting an appropriate risk analysis.

Contact plates with tryptone soya agar (IRR Tryptone Soya 1.6% Agar + Neutralizer N° 4, Redipor, Bicester, UK) were used for surface monitoring. This aspect of the environmental sampling included cleanroom windows, floors, walls, door handles, working surfaces (including laminar airflow chambers and tabletops) and devices (CO_2_ incubators, centrifuge, and microscope). After sampling, the surfaces were cleaned with 70% alcohol. The samples taken from the staff involved in the manufacturing process were also collected using this method. The samples taken from the garments and hands were collected at the end of each manufacturing day. For hand dabs, standard 90 mm diameter settle plates (IRR Tryptone Soya 1.6% Agar + Neutralizer N° 4, Redipor, UK) were used. The test was carried out at the same time schedule as in the passive air sampling method.

All of the collected samples from each of the methods were incubated for five days (with a possible extension of the incubation time by four days) at 35 °C ± 3. The results obtained from active air sampling are presented as the CFU/m^3^ of air. In the case of the settled plates, the results are reported as CFU/4 h, and the contact plates are reported as CFU/plate. In the case of positive samples from staff and grade A and B areas, the identification of contaminations was conducted in a specialized microbiological laboratory.

The correct cleanliness classes of the manufacturing rooms were confirmed at all stages as a part of the environmental control, and the total number of airborne particles (0.5 μm and 5 μm) was also measured according to the GMP rules- according to the ISO 14644-1:2015 standard by measuring during environmental control, referring to cleanrooms and associated controlled environments—Part 1: Classification of air cleanliness by particle concentration. No irregularities were noted.

### 2.3. Statistical Analysis

For statistical analysis, the chi-square test with Yates correction was used with the exception of the data presented in Table 4. For these results, Fisher’s exact test was used. The data were considered statistically significant at the level of significance *p* < 0.05.

## 3. Results

### 3.1. Detection of Microbial Contamination in the Biological Material

The results of the microbiological analysis of the biological material samples are shown in Table 1.

During the two-year clinical trial, 498 samples were collected for microbiological tests. The materials from all stages of ATMP manufacturing were studied, starting with fresh tumescent fluid, lipoaspirate, the fluids from two stages of cell isolation, and the primary cell culture. Overall, the numbers of samples taken from these manufacturing steps in the two consecutive years were 276 and 222, respectively. The contamination of samples taken from Stage 1 (fresh tumescent fluid) was detected in two probes (one each year). In Stage 2 (lipoaspirate), 41 samples derived from the two clinics were contaminated: 25/56 samples in the first year and 16/44 samples in the second year of the clinical trial. Contamination in the probes from Stage 3 (the medium from the isolated cells) and Stage 4 (the medium from the last wash of the cells) was still detected; overall, 12% and 6% of contaminated samples were found in Stage 3 and Stage 4, respectively. No contamination was detected in any probe collected from the subsequent primary cell culture (stage 5).

The cells obtained after processing 100 lipoaspirates were used to prepare 130 Advanced Therapy Investigational Medicinal Products and administered to the patients: 36 products were based on freshly isolated SVF cells, and 94 products contained expanded ADSCs (data not shown). The endotoxin results of the samples taken at the stage of the completed preparation of the medicinal product ranged between >0.05 and >0.25 EU/mL. All of the results obtained were below the accepted endotoxin level.

### 3.2. Identification of Microbial Contamination in Biological Material

Fresh tumescent fluid was contaminated by *Staphylococcus epidermidis*, *Staphylococcus capitis* and *Dermabacter hominis*.

More than 40% of the lipoaspirates were contaminated with thirteen different microorganisms. The most commonly isolated bacteria were *Staphylococcus epidermidis* (40%), *Propionibacterium acnes* (13%), *Staphylococcus capitis* (9%), and *Bacillus* spp. (9%) (Figure 2). Due to the difficulties in assigning *Bacillus* bacteria to individual species, all of the isolated bacteria of this genus are presented together in all figures.

Figure 3 shows the data of the microorganisms identified from the contaminated lipoaspirates depending on the incubation conditions of the collected samples (aerobic and anaerobic). Only five species were identified under both aerobic and anaerobic conditions: *Staphylococcus epidermidis*, *Propionibacterium acnes*, *Staphylococcus capitis*, *Staphylococcus hominis*, and *Staphylococcus lugdunensis*. In both cases, the most frequently identified microorganism was *Staphylococcus epidermidis*, which was isolated from over 50% of the contaminated samples. Under aerobic conditions, six other microorganisms were also isolated.

The bacteria identified in the samples from the subsequent stages of manufacturing (stages 2–4) are presented in Table 2. Microorganisms were detected in 40 samples of lipoaspirates (in eight samples, there was more than one bacterial species). In one case, despite a positive signal, no microbial growth was obtained. Bacterial contamination was detected in 12 samples taken from Stage 3, and these were the same bacteria as those identified in the lipoaspirate samples (Stage 2). In six cases, the contamination of the samples taken from Stage 4 was still associated with the original contamination of the probes from the lipoaspirates (Stage 2).

### 3.3. Environmental Monitoring of Microbiological Quality

During the two-year clinical trial, a total of 27,634 samples were collected under the environmental monitoring program at our GMP certified laboratory (details in Table 3), 22,173 of which were collected from Grade A and B areas. In this paper, only the results from routine aseptic monitoring of these GMP grades are presented. The types and number of the samples collected in areas of Grades A and B are shown in Table 4. During the two years, 15,644 samples were collected using the surface sampling method, 6393 samples were collected using the passive air sampling method (settle plates), and 136 samples were collected using the volumetric method. Comparing the results of the microbiological monitoring of the manufacturing environment between the first and second years of lipoaspirate processing, a decrease in the percentage of positive samples collected from the laboratory by passive air sampling and surface sampling was observed. For the settled plates method, this percentage was 2.00% in the first and 1.04% in the second year of the clinical trial, and 1.17% and 0.54% for the contact plates, respectively. Statistically significant differences were noted in the frequency of contaminated samples collected in the first and second years using these methods (*p* < 0.001 for surface sampling and *p* < 0.01 for passive air sampling). For active air sampling, we detected one contaminated sample each year.

The most commonly isolated and identified cleanroom bacteria were *Bacillus* spp. (44%), *Micrococcus* spp. (27%) and *Staphylococcus* spp. (19%). The remaining 10% included bacteria of other types (Figure 4). The analysis of the identified microorganisms from the samples collected using different methods revealed *Bacillus* spp. in over 60% of the positive samples obtained using surface sampling (data not shown).

A decrease in the percentage of positive microbiological results for the samples taken from staff directly involved in the manufacturing process was also observed, but the differences in the frequency of contaminated probes were not statistically significant. In the first year, microbiologically positive samples constituted 1.94% of all staff samples taken from both gloves and clothing, and in the second year, this percentage was 1.68% (data not shown). Figure 5 shows the data regarding the observed contamination for individual employees. Almost all employees (except one) reported a decrease in the number of positive tests in the second year. The most commonly isolated bacteria from the staff were *Bacillus* spp. (in 43.4%), *Staphylococcus* spp. (in 28.7%), and *Micrococcus* spp. (in 24.6%). Other identified species occasionally occurred, i.e., *Corynebacterium*, *Moraxella*, and *Kocuria* (Figure 6).

## 4. Discussion

MSC-based ATMPs hold great promise in relation to many unresolved medical problems [3,4,17]. Although classified as medicinal products, their method of production is not typical, so regulatory bodies try to look for special solutions for them, and scientists share their experience in solving manufacturing problems [25,28,32,33,34]. Our results concern the manufacturing of ADSCs, one of the most promising MSCs for cell-based therapies [10,13,35]. The most important finding is that, surprisingly, more than 40% of lipoaspirates, i.e., forty-one out of 100 samples received from the clinic to produce ADSC-based ATMPs, were contaminated. To the best of our knowledge, this is the first study presenting such an observation. It was an unexpected result when considering that the material was harvested by experienced clinicians that understood the entire procedure as participants in a clinical trial. The clinics were controlled by regulatory authorities, and above all, they were fully aware of the importance of the sterility of the collected material in view of the safety of the final product, as it is intended to be used in their patients. The implementation of the improved procedures in very determined and cooperating clinics brought about a decrease in the number of contaminated samples (from 44.6% to 36.4%) in the second year of the trial but did not eliminate them. Since all of the identified microorganisms are part of the physiological flora of human skin, the contamination is apparently related to the tissue donor site. This leads to the conclusion that the starting material for ADSC-based ATMPs should be considered contaminated even if it is harvested under the strictest regimen. Therefore, decontamination steps must be routinely implemented.

Our results document the successful implementation of the additional steps of the procedure, including the repeated rinsing of the starting material in an antibiotic and antimycotic solution. Other laboratories also use these or other antibiotics; for example, Golay et al. used gentamycin, showing a minimal amount of gentamycin in the final product after washing [36]. Martins et al. used an antibiotic-antimycotic solution during the initial stage of mesenchymal stem cell isolation from umbilical cord tissue but not in the cell culture medium [37]. Each cell laboratory develops its own strategy regarding the use of antibiotics-antimycotics at various stages of manufacturing based on the optimization of the type, dosage and duration of exposure, depending on the type of the source tissue, as well as the most frequent contamination type and level. The developed strategy must always be validated in terms of its influence on the quality and safety of the final products and should be based on individually adjusted risk analysis.

Microbiological monitoring during every stage of the procedure until the application, including the steps following manufacturing, should also be an issue of general concern for the manufacturer and the clinician. We have not had any such cases, but Vériter et al. reported contamination with *Staphylococcus aureus*, *Staphylococcus epidermidis* and *Corynebacterium* spp. in three out of nineteen samples of transport medium, revealed after the delivery of the ADSC-based products manufactured in their laboratory. The medium was sterile when leaving the manufacturing laboratory, so it must have been contaminated in the operating room [38]. Our results show the importance of sterility tests that were already performed at the stage of material acquisition in the clinic—in the case of ADSC—two extra samples, from tumescent fluid and lipoaspirate, are recommended to be controlled. As a consequence, not only the additional routine decontamination of tissue but also the individual adaptation of further steps, e.g., the duration of culture, may be applied. This is not only to extend the duration of exposure to antibiotics but to allow time for the microbiological results to be obtained.

A long wait for the final microbiological results is a known limitation. For the product, this is formally resolved through a two-step procedure, i.e., initial and final batch certification [28]. For manufacturing, we propose possibly extending the time in culture, with additional freezing if necessary, to determine need for additional decontamination depending on the results of the tests. A cost-benefit balance should be taken into account.

ATMP manufacturing takes place in cleanrooms of grades A, B, C and D, as defined by the *Rules Governing Medicinal Products in the European Union* (Volume 4) and the EU Guidelines on Good Manufacturing Practice Medicinal Products specific to Advanced Therapy Medicinal Products [28]. Not only the appropriate design of the manufacturing laboratory but also an environmental monitoring system is indispensable, and it should precisely define all activities related to microbiological environmental monitoring, i.e., the number of samples, the sampling technique and frequency, the acceptable limits on the number of microorganisms, and all of the actions to be performed in case of any microbiological incorrectness [39,40].

In this study, 22,173 samples from GMP Grade A and B areas were collected during a two-year clinical trial. Among all of these samples, only 248 (1.12%) showed bacterial or fungal growth. Passive air sampling results were positive for 102 agar plates, while volumetric air sampling revealed bacterial growth in only two samples (1.47%) taken from GMP Grade B. Interestingly, Martín et al., who reported the absence of microorganisms in all collected passive air samples and a definitely smaller number of positive samples from surfaces, observed bacterial growth in 38 samples (21.8%) and fungal growth in two samples (3.8%) collected via active air sampling in a Grade B area [41]. Tršan et al. also reported that most positive samples were collected with active air sampling [40]. Another study to assess the microbiological environment showed that approximately 47% of the air samples collected using volumetric analysis were free of microorganisms [42]. Differences in the data reported by various laboratories may be a result of the different sampling frequencies. The appropriateness of using the passive vs. active method is also currently under wider discussion; this topic is still waiting for a commonly accepted consensus [43,44,45,46]. For the microbial cleanliness of the laboratory surfaces, Cobo and Concha reported a large amount of floor surface contamination, especially near critical equipment such as incubators and centrifuges, as well as a significant amount of wall contamination [43]. In our laboratory, most cases of contamination were found in relation to cleanroom windows and door handles (Table 4). Most importantly, we achieved a significant improvement in the second year of production in both cases, where we did not find any contamination of the working surfaces. This confirms the importance of continuous staff training as well as a constant verification of staff behavior patterns in clean rooms.

The predominant bacterial genus isolated in our study was *Bacillus* spp. (44%), *Micrococcus* spp. (27%), and *Staphylococcus* spp. (19%). Other bacterial genera accounted for 10% of all of the identified microorganisms. In previous studies, the most commonly isolated microorganism was *Staphylococcus* spp. [40,41,43] or *Micrococcus* spp. [47]. The authors also showed the presence of bacteria from the *Bacillus* spp. genus.

Sandle [47] noted that 97% of the bacteria recovered from 40 Grade A and B cleanrooms and clean zones were Gram-positive. In this study, the vast majority of identified microorganisms were also Gram-positive bacteria, which is in agreement with the results reported by Cobo and Concha [43] and Martín et al. [41]. Gram-positive rods (for example, *Bacillus*) are usually transferred to cleanrooms on equipment or dust. Gram-positive cocci (such as *Staphylococcus* spp. and *Micrococcus* spp.) are usually associated with a normal human microbiota and therefore often occur in clean rooms [47]. The bacterial species isolated in this study are usually nonpathogenic. Nevertheless, any contamination of ATMP entails the disqualification of the final product in accordance with GMP requirements. Therefore, to eliminate any risk of contamination of the product, continuous improvement of the procedures used, including the sanitization program for the cleanroom, is required.

More than half of the microorganisms that cause contamination in cleanrooms for manufacturing aseptic products originate from the normal skin flora of the staff. Microorganisms can be isolated from staff working in cleanrooms, where full-body covers with hoods and masks are worn [43,47]. Therefore, sampling from staff is another important element in the assessment of microbiological quality control. In this study, samples were collected from each staff member in a cleanroom after working with the cells of each donor. The most commonly isolated microorganisms were *Bacillus* spp. (43.4%), *Staphylococcus* spp. (28.7%), and *Micrococcus* spp. (24.6%), coinciding with the most commonly identified microorganisms in the monitored cleanroom environment. The percentage of positive samples decreased in the second year of manufacturing (1.68% compared to 1.94%). This is undoubtedly due to the acquisition of more staff experience and training during the manufacturing period.

Lastly, it should be emphasized that even if a well-designed contamination control strategy is in place with regard to ATMPs that contain cells derived directly from the culture, the batch release is performed prior to obtaining all QC results. Faster, reliable microbial tests would be highly desirable, as they would significantly improve the safety of the patients receiving ATMPs. Given the development of cell-containing products with a short shelf life, there has been an intensive development of rapid microbiological methods (RMM) [48,49,50,51] accepted by regulatory bodies as so-called “alternative methods” [31,52]. However, for the products with shelf lives of a few hours, currently available RMMs which shorten the test from the classical 14 days to 7 days are still too long.

To conclude, for ADSC-based ATMPs, the starting material should be considered contaminated. However, that does not prevent the obtaining of a sterile final product for release from the clinic as long as appropriate actions are taken. Additional routine decontamination steps should be included at the very beginning of manufacturing. Microbiological control at the stage of material acquisition in the clinic is indispensable, and the results should be taken into account in terms of further manufacturing steps. The proficiency of personnel in the conditions of a specific manufacturing process increases the quality of the production environment. Regardless of these specific implications, the results presented support the common voice of the community for the further development of rapid microbiological tests.

## Figures and Tables

**Figure 1 cells-12-00680-f001:**
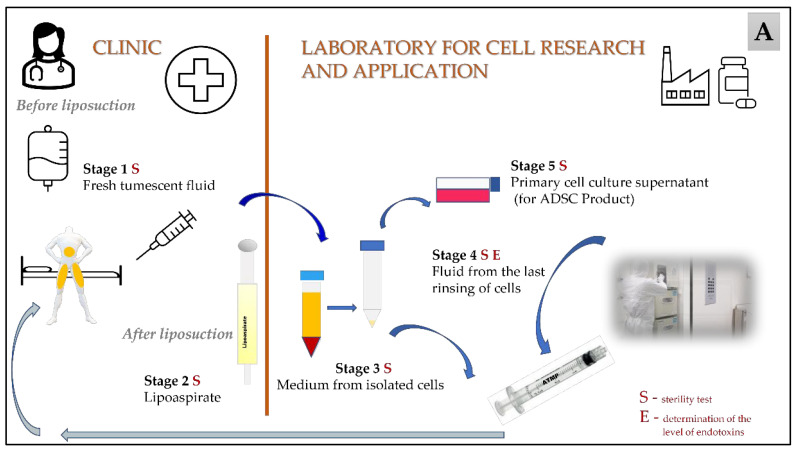
Subsequent stages of the procedure, from the collection of the starting material to the production of ATIMP; (**A**) overall schema of the whole process; (**B**) detailed manufacturing steps in the laboratory. The sampling points for sterility and endotoxin tests are marked with the letters S and E, respectively.

**Figure 2 cells-12-00680-f002:**
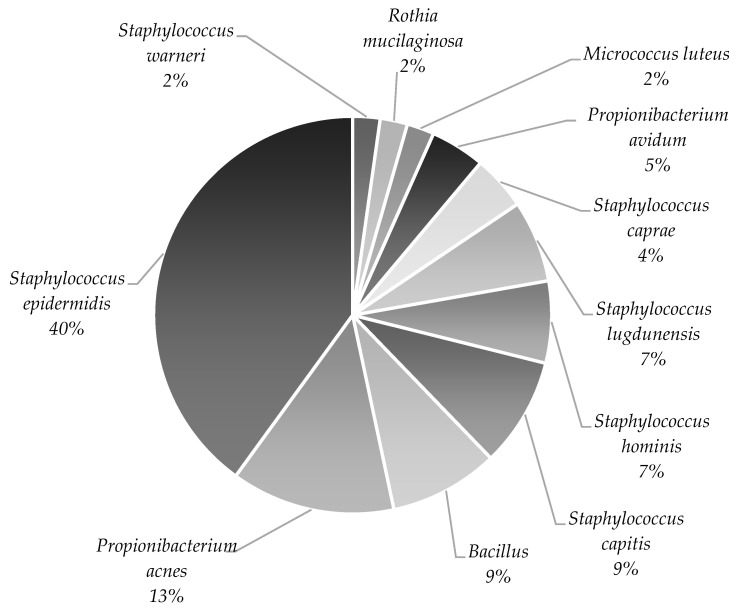
Microbiota identified in lipoaspirates collected throughout the clinical trial are presented as the % of each genus out of all the identified microorganisms. All isolated bacteria of the *Bacillus* genus are presented together.

**Figure 3 cells-12-00680-f003:**
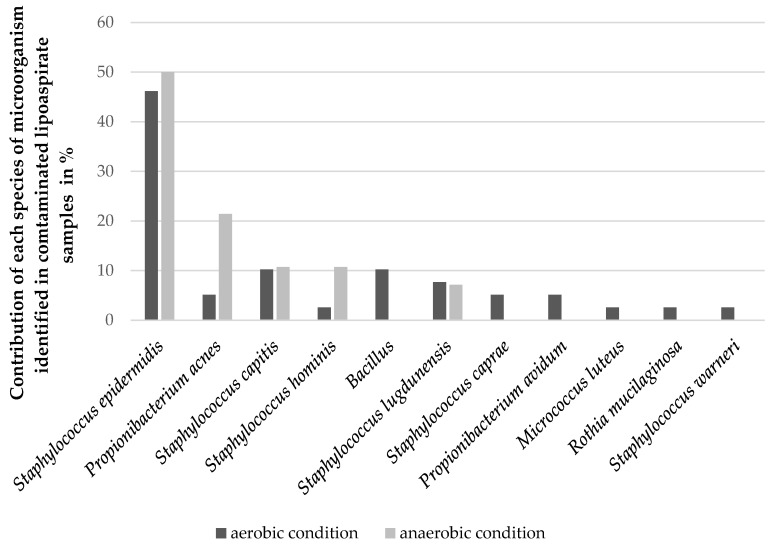
The frequency of individual microorganisms isolated from contaminated lipoaspirates identified after incubation under aerobic and anaerobic conditions. All isolated bacteria of the *Bacillus* genus are presented together.

**Figure 4 cells-12-00680-f004:**
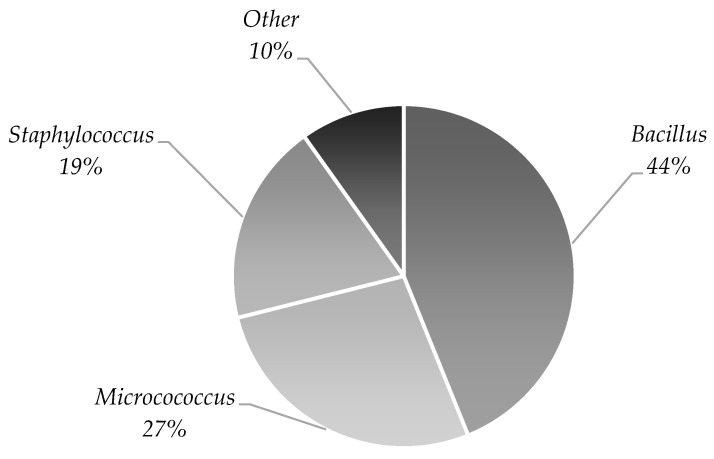
Cleanroom microbiota identified during the two-year clinical trial presented as % of isolated genera out of all identified microorganisms.

**Figure 5 cells-12-00680-f005:**
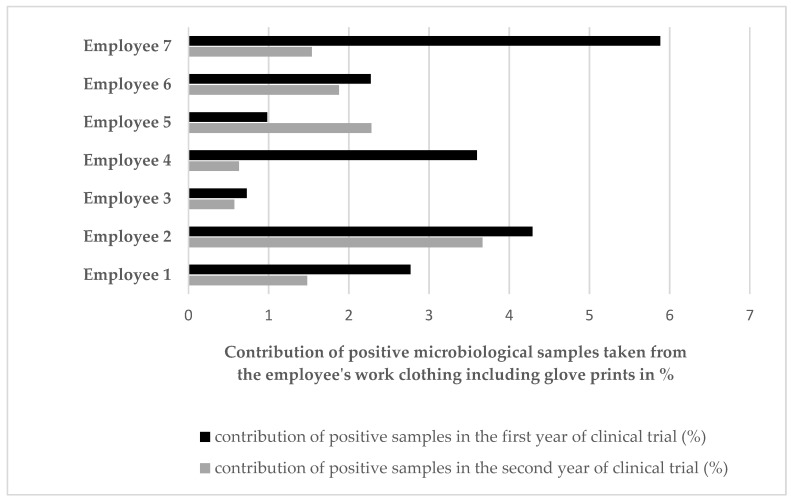
Microbial contamination observed in the samples taken from the staff (including glove prints). The diagram shows the total score for each individual employee by year over the course of the entire trial.

**Figure 6 cells-12-00680-f006:**
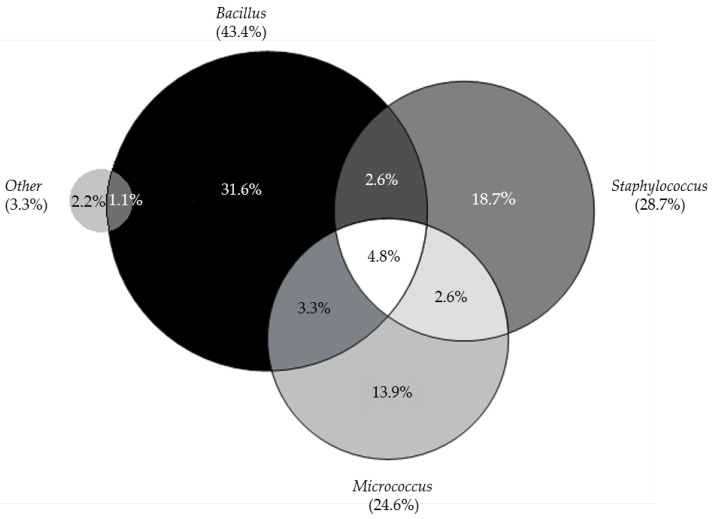
Microbial flora identified in the samples taken from the staff (glove prints and work clothing) during the entire trial. The circles representing individual genera are proportional to the frequency of occurrence (indicated in brackets), presented as % of each type of identification out of all identified samples.

**Table 1 cells-12-00680-t001:** Contamination of the processed material at subsequent stages—from harvesting of the adipose tissue to the primary cell culture—observation from a two-year clinical trial.

Year of Clinical Trial	Total Number of Samples	Stage 1 Fresh Tumescent Fluid		Stage 2 Lipoaspirate		Stage 3 Medium from the Isolated Cells		Stage 4 Medium from the Last Wash of Cells		Stage 5 Cell Culture Supernatant	
		Number of Samples	Positive/% Contaminated	Number of Samples	Positive/% Contaminated	Number of Samples	Positive/% Contaminated	Number of Samples	Positive/% Contaminated	Number of Samples	Positive/% Contaminated
Year 1	276	56	1/1.8	56	25/44.6	56	8/14.3	56	3/5.4	52	0/0
Year 2	222	44	1/2.3	44	16/36.4	44	4/9.1	44	3/6.8	46	0/0
Total	498	100	2	100	41	100	12	100	6	98	0

**Table 2 cells-12-00680-t002:** The number of samples contaminated with specific microorganisms in the subsequent stages of the isolation process. The data include the results of all samples from stages 2 to 4 collected throughout the clinical trial. A single probe could be contaminated with more than one species. All isolated bacteria of the *Bacillus* genus are presented together. Empty fields listed in the table represent that no samples were contaminated with microorganisms for that particular stage of the isolation process.

Microorganism	Stage		
	Stage 2 Lipoaspirate	Stage 3 Medium from the Isolated Cells	Stage 4 Medium from the Last Cell Wash
*Staphylococcus epidermidis*	22	5	2
*Propionibacterium acnes*	6	4	2
*Bacillus*	4		
*Staphylococcus capitis*	4	1	1
*Staphylococcus hominis*	4		
*Staphylococcus lugdunensis*	3	1	
*Staphylococcus caprae*	2	1	
*Propionibacterium avidum*	2	1	1
*Micrococcus luteus*	1		
*Rothia mucilaginosa*	1		
*Staphylococcus warneri*	1		

**Table 3 cells-12-00680-t003:** The number of samples taken from cleanroom environments with different sampling methods during a two-year clinical trial.

Year of Manufacturing for Clinical Trial	Number of Samples Collected from All Monitored Grade Areas		
	Surface Sampling (Contact Plates)	Passive Air Sampling (Settle Plates)	Active Air Sampling (Volumetric Sampling)
Year 1	11,670	4731	128
Year 2	7588	3389	128
Total	19,258	8120	256

**Table 4 cells-12-00680-t004:** Data of positive samples from Grade A and B areas taken with different sampling methods. Statistically significant differences in the frequency of contamination in individual years are marked with square brackets.

Sampling Surface	GMP Grade	Two-Year Manufacturing Period						Differences between Years
		Year One			Year Two			
		Total Number of Samples	Number of Positive Samples	% of Positive Samples	Total Number of Samples	Number of Positive Samples	% of Positive Samples	Statistical Significance
Cleanroom windows	B	307	21	6.84	417	12	2.88	*p* < 0.05
Floor	B	2200	45	2.05	1498	18	1.20	ns
Door handles	B	1300	30	2.31	875	3	0.34	*p* < 0.001
Walls	B	1277	5	0.39	563	1	0.18	ns
Devices	B	1438	5	0.35	1005	1	0.10	ns
Working surfaces	A + B	2863	5	0.17	1901	0	0.00	ns
Laminar air flow chamber	A	2708	3	0.11	1798	0	0.00	ns
Tables	B	155	2	1.29	103	0	0.00	ns

## Data Availability

All data generated or analyzed during this study are included in the published article.

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
