# Peer review of "Microbiological Aspects of Pharmaceutical Manufacturing of Adipose-Derived Stem Cell-Based Medicinal Products"

_cells, 2023, doi:10.3390/cells12050680_

Round 1
Reviewer 1 Report
The manuscript by Gadomska et al demonstrated Microbiological Aspects of Pharmaceutical Manufacturing of Adipose-Derived Stem Cell-Based Medicinal Products. The authors first examined the sterility of biological material and studied the microbiological aspects. The presented results and conclusions have a wider clinical significance, due to the very intensive development of various therapies with the use of adipose-derived stem cell-based medicinal products in many different diseases. However, the authors should address the following comments.
1. Even though highly beneficial effects of MSCs transplantation; why FDA did not approve MSC transplantation clinically? What are the limitations of MSCs from the translational perspective? Authors should discuss this.
2. Authors should discuss the precautionary methods that could eliminate microbial contamination in cleanroom practices.
3. How to perform quality control for high-risk raw biological materials
4. Authors should include mycoplasma cross-contamination testing prior to cell harvest.
5. Authors should update with the latest references
Author Response
Dear Sir, Madame,
Thank you very much for your opinion on our manuscript.
Please, find the detailed response to your comments in the attached file.
Sincerely yours,
Malgorzata Lewandowska-Szumiel

Reviewer 2 Report
In the abstract, the word "specificity" is causing ambiguity (Line 17).
Authors should use a conjugation at the start of the 3rd sentence (Line 19)
The term “well-designed manufacturing protocols can be replaced with good manufacturing practices. (Line 28)
The last sentence of the abstract is non-conclusive. Authors should mention why well-designed manufacturing protocols etc., are required.
In the introduction section, the authors have given a long account of the role and mode of action of mesenchymal stem cells, while the topic of the research article is related to the production and quality control of stem cells. In this scenario, the introduction contains unnecessary information and detail. It is highly recommended to rewrite the introduction considering the isolation, manufacturing, and quality control of stem cell production. Additionally, the authors have not provided any reference to their claims (Lines 78-83)
The authors claim that study was carried out for 2 years, while the clinical trial data showed the study duration was only 1 year. Can you please explain this?
What were the experimental parameters for preparing stroll vascular fraction? (Line 103-104).
Lines 107-110 are not making any sense; please rewrite them.
In line 114 of section 2.1 authors mention 4 stages and later provide 5. Please clarify this.
What type of validation author are you referring to? (Line 128) Moreover, the authors have only chosen six strains for their study.
The term “sample environment” needs an explanation. (Line 131)
A laminar airflow chamber without HEPA filters can easily be a source of microbiological contamination; instead, a certified biosafety cabinet is recommended for this investigation. Why did the authors use a laminar flow chamber?
Why the word “Pathogen” is used instead of a contaminant? (Line 140). How was the pathogen identification performed? Please give a detailed account.
Kindly provide the reference for lines 143-144.
Figure 1 is poorly presented, and the labeling is inappropriate, while in the 1B the reader can not understand from which side the smart art is starting.
Please provide the details of the “approved microbiological monitoring program” (Line 158)
Why were the samples not collected at the start of the day/manufacturing process? (Line 170-171)
The units require correction (Line 177)
The names provided in Figures 2 and 3 need to be italicized.
Please explain and provide data for “others” (Line 293)
The first part of the discussion lacks scientific citations and agreement with previous or related published articles, which is the essence of scientific communication. Moreover, it seems the reputation of the results. (Line 306-363)
The discussion part is too long. It should be to the point.
Authors must also provide a concise conclusion, including the implications of their findings.
Author Response

(The authors gave the same response as above.)

Round 2
Reviewer 1 Report
Accept
Author Response
Dear Sir/Madame,
Thank you for the positive opinion.
Sincerely Yours,
Malgorzata Lewandowska-Szumiel
Reviewer 2 Report
The authors have subsequently improved the quality of the text. However, the illustrations/flow chart in Figure 1, they presented could be of better quality and standard. It is highly recommended to use some open-access websites for illustration re-designing. Attractive and concise illustrations are the essence of modern scientific publications. In this way, the authors can develop the interest of the reader and can convey their findings effectively.
The authors should include the error bars in the graph of Figures 3, and 5, as they already mentioned that they performed the experiments multiple times.
There are several places where the font style could be more consistent. Authors must follow the “Cells” recommended font style in the text, figures/graphs, etc.
Line spacing in tables 1~4 is extraordinary; it needs to be fixed.
Author Response
Dear Sir/Madame,
Thank you very much for your comments.
Detailed responses to each of your remarks can be found in the attached file.
The revised manuscript has been uploaded as well.
Sincerely Yours,
Malgorzata Lewandowska-Szumiel
